# Novel Probe for Thermally Controlled Raman Spectroscopy Using Online IR Sensing and Emissivity Measurements

**DOI:** 10.3390/s22072680

**Published:** 2022-03-31

**Authors:** Chiara Calvagna, Andrea Azelio Mencaglia, Iacopo Osticioli, Daniele Ciofini, Salvatore Siano

**Affiliations:** Istituto di Fisica Applicata “Nello Carrara”-Consiglio Nazionale delle Ricerche (IFAC-CNR), 50019 Florence, Italy; c.calvagna@ifac.cnr.it (C.C.); a.mencaglia@ifac.cnr.it (A.A.M.); i.osticioli@ifac.cnr.it (I.O.); d.ciofini@ifac.cnr.it (D.C.)

**Keywords:** laser heating, emissivity, Raman spectroscopy, pigment, IR sensor

## Abstract

Temperature rise during Raman spectroscopy can induce chemical alterations of the material under analysis and seriously affect its characterization. Thus, such photothermal side effects can represent a serious problem to be carefully controlled in order to safeguard the integrity of the material and its spectral features. In this work, an innovative probe for thermally controlled portable Raman spectroscopy (exc. 785 nm) equipped with infrared sensing lines was developed. It included an infrared source and two thermopile sensors, which allowed to perform real-time measurements of the local emissivity of the material surface under laser excitation. The emissivity, which is needed in order to monitor the temperature of the irradiated surface through infrared radiation measurements, represents the complementary component of the reflectance in the radiative energy balance. Thus, total reflectance, temperature measurements and Raman spectroscopy were integrated in the present probe. After independently assessing the reliability of the former in order to derive the emissivity of variety of materials, the probe was successfully applied on pigments, paint layers, and a painting on canvas. The results achieved evidence the significant exploitation potential of the novel tool.

## 1. Introduction

Raman spectroscopy is a widespread technique in archaeometry, conservation of cultural heritage (see for example) [1], and many other fields. In particular, it is often used to analyze paint layers in order to collect information on the execution techniques, authenticity, and the state of conservation of a given artefact, as well as to define and assess the result of conservation treatments, such as those based on laser ablation [2,3]. However, extensive applications on easel and mural painting including many measurements are rarely carried out because of intrinsic risks of damage associate to the high operative laser intensities and total radiant exposures.

Although Raman spectroscopy is generally considered a non-destructive technique, laser heating represents a crucial issue for materials with high optical absorption at the excitation wavelength and low critical temperatures. Laser heating may produce band broadening and shifting, background increase, and even sample damage [4]. Phase transitions, discolorations, or photo-degradation of the samples can be induced by laser irradiation at the typical intensities and total radiant exposure of Raman spectroscopy [5,6,7]. The measurement and control of the laser induced temperature rise can allow avoiding data misinterpretation and undesired alterations.

It is well known that temperature affects the distribution of the population of the ground and excited vibrational states, so it can be found out from the ratio of Stokes and anti-Stokes lines intensities [8,9]. However, the anti-Stokes bands for many materials are very weak so it is not trivial to achieve reliable temperature using their intensities. Moreover, this method is expensive because it requires the use of a Super-Notch filter and it is not suitable for in situ measurements [10,11].

In the present work, we introduce a novel, non-contact temperature monitoring and then control method based on the measurement of the surface emissivity and its infrared (IR) emission. Emissivity (*ε*) is defined as the ratio of the radiant emittance of the sample and that of the black body, as recorded at the same temperature. It is a dimensionless number and it varies from zero (perfect reflector) to one (black body). This parameter, which represents the efficiency with which a material radiates the absorbed energy, must be known or preliminarily determined whenever approaching non-contact temperature measurements based on IR sensors [12,13,14].

Various setups and instruments have been developed and demonstrated, which allow measuring the emissivity. The main experimental techniques used are based on calorimetry [15,16], radiometry [17,18], or thermography [19,20,21], usually referred to as direct methods, and on the measurement of spectral reflectance, usually referred to as indirect method [12,22]. The latter is straightly derived from the energy balance for a diffusing sample under irradiation: *E*_i_ = *E*_a_ + *E*_R_ + *E*_T_. Since the absorbed energy, *E*_a_, equals the emitted energy, *E*_e_, (*E*_T_ and *E*_R_ do not contribute to the latter), the energy balance can be rewritten as *E*_i_ = *E*_e_ + *E*_R_ + *E*_T_ or also as 1 = *ε* + *R* + *T*, where *R* and *T* are the reflectance and transmittance, respectively. Furthermore, *T* is negligible in the IR region for most solid samples, thus usually: 1 = *ε* + *R*, which corresponds to the complementarity between *ε* and *R* (emissivity increases when the reflectance decreases).

In this work, we report an innovative IR sensing setup allowing for carefully mapping the emissivity of a material surface, as derived from reflectance measurements, and then to monitor and control the temperature rise during compositional mapping using Raman spectroscopy. The setup included an integrating sphere with four radiation channels for the measurement of the total reflectance by means of IR radiation source and sensor channels, of the thermal radiation from the laser excited area, and of the Raman spectrum, respectively. The demonstration of its effectiveness and reliability along with the small sizes of its components make the thermal control channels here developed suitable for integration with a variety of portable as well as laboratory Raman instruments. Furthermore, such an approach to the measurement of the CW laser induced heating could also be useful in laser material processing and other.

## 2. Experimental Set-Up

### 2.1. Emissivity Measurement

Different materials of known emissivity (*ε*) ranges were selected, according to the literature data [23,24,25,26,27], in order to cover the whole interval between 0–1 through suitable sample preparation phases. The set of materials included metals (Al, Cu, Pb, Fe, and steel with different surface textures), plywood, gypsum, glass, marble, paper, PVC tape, and limestone.

The respective values of *ε*, which depends on the specific surface features, were directly measured using a setup including Peltier cell, digital contact-thermometer equipped with K type thermocouple (Fluke, Seaway, WA, USA), and indirectly derived through an IR sensor based on a thermopile (TP_R_) with detection range 6–14 µm (Phidgets Inc., Calgary, AB, Canada). The latter is provided with a blackbody calibration (*ε* = 1), digital reader, and app allowing to set any emissivity value between 0–1 and to plot the temperature time evolution.

Each sample was then placed on a small aluminum slab in strict thermal contact with the heated face (76 °C) of the Peltier cell and a thermocouple, which measured its temperature. When the sample reached the thermal equilibrium, its temperature was registered and, simultaneously, the corresponding IR emission was detected by the TP_T_ sensor (a thermopile similar to TP_R_). Thus, the emissivity of the sample surface was derived by suitably setting its value in the app of the sensor in order to achieve the same temperature as that directly measured by the thermocouple.

Following several preliminary tests, a probe terminated by an integrating sphere equipped with four radiation channels and allowing for indirect emissivity and temperature measurements along with Raman spectroscopy was developed (Figure 1). The internal surface of the sphere was coated with a reflecting aluminum film providing a homogeneous distribution of the IR radiation within the sphere itself during the measurement of the reflectance, *R*, of the samples under investigation. The latter was carried out using a partially collimated IR source (EOT, Pordenone, ITA) focused onto the sample surface through a ZnSe lens L_1_ (*f*_1_ = 15 mm) and the thermopile sensor, TP_R_.

The IR emission produced by the laser heating of the sample during Raman excitation was collected in a confocal configuration and imaged onto the active area of the thermopile, TP_T_ (same specs as those of TP_R_) using a ZnSe lens L_2_ (*f*_2_ = 15 mm, magnification factor = 3).

A LabVIEW™ virtual machine was developed for controlling the IR source and the thermopile sensors, deriving the reflectance and emissivity values, and performing Raman spectroscopy through a dedicated graphic user interface (GUI). To derive the total reflectance, *R*, the IR radiation scattered by the sample was compared with that scattered by a Cu sample, as derived through the contact thermal measurement described above: *R_Cu_* = 1 − *ε**_Cu_*. The sample temperature was detected in both IR source ON (*T*) and OFF (*T*_0_) conditions for removing the room temperature background. Moreover, the thermal contribution of the sphere, as measured with the IR source switched ON without the target (background component due to the radiation scattering of the entrance and exit apertures) was subtracted (*T_sp_*). *R* was hence calculated according to the following formula:(1)$$R=\frac{T^{4}-\left(T_{0}^{4}+T_{sp}{}^{4}\right)}{T_{Cu}^{4}-\left(T_{0}^{4}+T_{sp}{}^{4}\right)}R_{Cu},$$
where *T_Cu_* is the recorded temperature for the Cu reference sample and *R_Cu_* is the corresponding reflectance, while the fourth powers of the temperatures come from the relation between the emittance and the temperature given by the Stefan-Boltzmann law. Equation (1) allowed to indirectly derive the emissivity as *ε_R_ =* 1 − *R.*

### 2.2. Temperature Control

The Raman system was a commercial fiber coupled portable spectrometer (i-Raman, B&W Tek, Township, NJ, USA) extensively used in archaeometry and conservation [28,29]. Its laser excitation beam (785 nm) was focused onto the sample at an angle of 30° (with respect to the normal) through a plano-convex lens L_3_ (Figure 1) with a focal length *f_3_* = 20 mm. The scattered light was collected through the same lens and sent to a spectrometer providing a spectral resolution of about 8 cm^−1^.

The sizes of the focal spot were measured using a digital microscope (Dino-Lite, AnMo Electronics Co., New Taipei City, Taiwan) at high magnification (500×). The laser spot showed an elliptical shape (Figure 2a) with enough regular transverse (blue) and longitudinal (black) profiles (Figure 2b), respectively.

The best fit of the curve was achieved through a super-Gaussian function with index 1.2 (Figure 2b). The semi-major and semi-minor axis corresponding to 77% of the total area were about 215 and 168 µm, respectively. Thus, the mean diameter of the focal spot was about 383 µm.

The temperature was calculated using the following formula derived from Stefan-Boltzmann black body law (see for example [30]) by taking into account the environmental contribution and the fact that the laser spot covered only a part of the field of view (FoV) area [6]:(2)$$T=\sqrt[{4}]{\frac{T_{s}^{4}-\left(1-\varepsilon \right)\cdot R_{A}\cdot T_{0}^{4}\cdot \theta -\left(1-R_{A}\right)\cdot T_{0}^{4}}{\varepsilon \cdot R_{A}\cdot \theta }}$$
where *T_s_* is the apparent blackbody temperature (*ε* = 1) provided by the thermopile sensor TP_T_, *R_A_* = laser spot area/FoV area, and *θ* the lens transmissivity (0.95 in the present setup).

Equation (2) represents the recalibration of the thermal sensor according to the specific setup and material understudy, whose emissivity, *ε*, was measured as described above. The areal ratio *R_A_* in Equation (2) was experimentally determined by moving a thin metallic wire across the laser spot, along the *Y* and *X* axis, respectively. The wire was heated at a constant temperature (70 °C) with a suitable electric current and for each position, two temperatures were measured: with the laser switched OFF and ON, respectively. The detected temperature rises Δ*T*(*X*) and Δ*T*(*Y*) with the laser OFF shown in Figure 3 (blue and black lines, respectively) provided the FoV of the sensor TP_T_. The same scans with the laser ON (red and green lines, respectively) provided an estimation of the diameter of the wire (180 µm) and an alternative measurement of the diameter of the laser focal spot (about 380 µm). Furthermore, the measurement also allowed to verify the good alignment of the sensor TP_T_.

By considering the area of the laser spot, as measured using the beam analyzer (Figure 2), and the FoV area, as derived from FWHMs of the curves of Figure 3, the areal ratio to be used in Equation (2) was *R_A_* = 0.14.

## 3. Results

As mentioned above, various material samples were formerly investigated. The measurement of their emissivity according to the direct (*ε**_T_*) and indirect (*ε**_R_*) method, respectively, are listed in Table 1, along with some related literature data (*ε**_ref_*).

As shown, an overall satisfactory agreement was achieved, which evidenced, in particular, the reliability of the indirect method using the integrating sphere shown in the inset of Figure 1. This is also clearly evidenced in Figure 4, showing the congruence with the physical relation between reflectance and emissivity in the approximation of opaque materials [23]. Thus, such a comparison demonstrated the functionality and reliability of the setup of Figure 1 and of the associated reflectance method for the samples listed in Table 1.

As mentioned above, one of the main potential applications of the present innovative thermally controlled Raman probe concerns the field of archaeometry and conservation of cultural heritage. Thus, following the assessment of the effectiveness of the present thermal monitoring approach, the novel probe (Figure 1) was tested on a set of pigments in linseed oil binder applied on glass slides. The samples and the corresponding emissivity values, as measured using the direct method (*ε_T_*) or the indirect method based on reflectance (*ε**_R_*), are reported in Table 2.

The comparison confirmed the congruence between the two independent methods, with maximum differences of about 0.1 for pigment and a bit less for paint layers. The emissivity of the painted samples was generally higher than that of the corresponding pigment because of the higher emissivity of the linseed oil matrix. The pigment: binder ratio was 65 wt%:35 wt%, which corresponds to a pigment volume concentration fraction of 17–30% for the actual pigments. Thus, a certain variation is expected in practical cases of interest, due to the different ratio between pigment and binder, as well as to the presence of two or more pigments in the paint mix.

The emissivity measured for the various samples was used to recalibrate the temperature sensor TP_T_ and implement the temperature control during Raman spectroscopy. As an example, in Figure 5, the case of cinnabar, a red pigment with high photothermal instability, is reported. Figure 5a displays the Raman spectra achieved at eight different laser powers with an integration time of 12 s and Figure 5b shows the respective temperature profiles.

The Raman signal gradually increased up to 86 mW (yellow line), whereas above this power, drastically decreased (dark yellow line). In addition to the previous spectral feature, even a significant variation in the temperature temporal profile was observed at laser power higher than 86 mW. In particular, at 88 mW, the slope of the temperature profile changed drastically at 11 s, and 1 s later it increased from about 200 °C to about 325 °C. In the inset of Figure 5b, the temperature recorded at 11.6 s as a function of the laser power is reported. Temperature values were very reproducible, as assed through 30 consecutive measurements, which returned a standard deviation of 1–2 °C (within the square symbol of Figure 5). As shown, the linear increasing trend had an abrupt change of slope at 86 mW (160 °C). This variation was associated with the darkening of the irradiated area. According to the literature [31], such a darkening effect and associated metallic appearance were attributed to a photothermal reduction effect occurring at the vaporization of mercury (357 °C). The corresponding experimental critical temperature was lower than the nominal one likely because of the fast rise, which cannot be detected by the sensor (response time of 1.3 s) and of the associated material removal, which produce a cooling effect [6]. Nevertheless, the present thermal sensing line allowed to determine quite precisely the alteration threshold, as well as to foresee the damage through the lack of saturation of the temperature profile. Thus, in all the cases the maximum temperature of the irradiated sample surface along with a maximum slope of the temperature temporal profile, after its former leading edge, can represent two control parameters that can be suitably set through the dedicated GUI in order to optimize the signal detection and prevent possible alterations.

As highlighted in the literature, the alteration intensity strongly depends on the laser spot diameter [5,32]. For example, we previously found about 600 W/cm^2^ for a spot size of 105 µm, against the present 76 W/cm^2^ for about 380 µm spot diameter. This general behavior should be taken into account along with other features of the laser damage nucleation, which extends beyond the subject of this paper.

Critical temperature and alteration intensities of a paint layer are expected to be significantly lower than those of the inorganic pigments that it contains, because of the thermal sensitivity of the organic components (binder, dye), but the conceptual approach of the thermal control described above remains the same: suitable constrains to the thermal temporal profile can prevent sample alteration.

Figure 6a shows the spectra achieved at different laser emission power on a 10-years aged paint sample prepared with cinnabar in linseed oil on gypsum primed wooden panel. Raman spectra were achieved with an integration time of 12 s (100 ms, 120 averages). In Figure 6b the respective temperature profiles are reported. The temperature rise induced by the laser irradiation increased linearly with the laser power (see the inset) and the sample didn’t show any alteration. This can be attributed to the higher optical penetration depth of the paint film with respect to the pure pigment.

A further application of the present innovative probe was carried out on Prussian blue, which is another known photothermally sensitive pigment. It is an iron (III) hexacyanoferrate (II) complex and its color is correlated with the iron (II) and iron (III) charge transfer via a cyano group.

Figure 7 shows the dependence of the Raman spectrum (integration time of 65 s) and associated temperature profile on laser power of Prussian blue in linseed oil paint film. No alteration was observed over the indicated power range (27–103 mW). In addition to band and background amplitude variations increasing red shift of the CN stretching band (≈2150 cm^−1^) was observed at increasing laser power. This is clearly observable in the detail of Figure 8a showing the baseline subtracted spectra between 2105–2180 cm^−1^. No other shift was observed. A Gaussian fit of the mentioned band was executed in order to extract its central wavenumber for each spectrum. A red shift of 4 cm^−1^ was observed when changing the irradiation power from 27 mW to 103 mW laser power. Such an effect was previously reported [33,34] and it was attributed to the reduction of Fe (III) to Fe (II) in absence of charge transfer process. Here, we correlate it to the laser induced temperature rise (Figure 8b).

In order to validate the novel probe in real operative conditions tests were carried out on a modern oil painting on canvas. It was a female portrait by anonymous, which was overpainted and partially uncovered in the past at its bottom part (Figure 9). The original painting (below the dotted line) was extensively overpainted using green, yellow, and brown hue paint. This artwork was previously analyzed using optical and ESEM-EDX microscopy and uncovering laser ablation tests were carried out [35].

For the present purposes, the Raman spectra along with emissivity and temperature measurements were achieved on the reddish dress (sites A and B) and the green overpaint (site C). After some preliminary tests, the laser power was set at 47 mW and the integration time at 25 s (500 ms, 50 averages) for the site A (Figure 10a) and 22 mW and 80 s (200 ms, 400 averages) for site B (Figure 10b). The intensity of the latter measurement was reduced since the temperature induced by the former has to be considered too high for safeguarding the oil binder. The emissivity was measured in situ and resulted to be 0.91 for site A and 0.87 for site B. Site A (orange zone) shows the red lead characteristic peaks (222, 306, 386, and 541 cm^−1^), and in site B (red zone) besides these the Raman bands of the quinacridone (1273, 1356, 1382, 1486, and 1576 cm^−1^) were also detected.

The spectra and temperature rise of the green overpaint (site C) were finally measured using a laser power of 22 mW (Figure 11) and an integration time of 40 s (800 ms, 50 averages). The Raman characteristic bands of the green phthalocyanine were recognized (680, 738, 1208, 1286, and 1532 cm^−1^). The Signal/Noise level of this spectrum could be raised up using higher laser power, although the induced temperature rises could not be compatible with such a fully organic site (organic pigment and binder).

These examples show the monitoring of the temperature allowed to evaluate and then control the level of risk of the measurement. In particular, literature data report the evaporation of VOCs contained in oil matrix represent the early alteration of oil paint, which starts to occur over a large temperature range around 100 °C, according to the specify features. Thus, if this temperature has to be reached in order to acquire a good spectrum, the time exposure should be limited. The measurements carried out on the painting of Figure 9 involved maximum temperature rises between 75–90 °C, which did not evidence any anomalous behavior, although it would be safer to further limit the heating.

## 4. Conclusions

Here, we extended the technology and method of the remote temperature control during Raman spectroscopy previously introduced. A novel portable probe was developed allowing to derive the emissivity from reflectance measurement and then to use this parameter to monitor the temperature of the irradiated sample surface during Raman spectroscopy. The effectiveness and reliability of the indirect measurement of the emissivity based on IR reflectance measurement was formerly demonstrated on a variety of material samples including metals, stones, and other, the work focused on pigments and paint layers. The latter were selected as an example application, where the thermal control during Raman spectroscopy can represent the solution of a practical need because of the high photothermal sensitivity of most of the pigments and binders typically encountered in artwork material characterizations. These material systems easily undergo alteration during Raman spectroscopy, which can be prevented by implementing an online temperature control. At the same time, it is evident that the present approach can also be exploited in other fields of application, whenever the knowledge of the temperature during CW laser irradiation can be of interest. One case of particular interest is the study of the dependence of the Raman spectral features on the temperature, such as for example the red shift of Prussian blue discussed above, where the contribution of the laser irradiation should be carefully taken into account and measured.

## Figures and Tables

**Figure 1 sensors-22-02680-f001:**
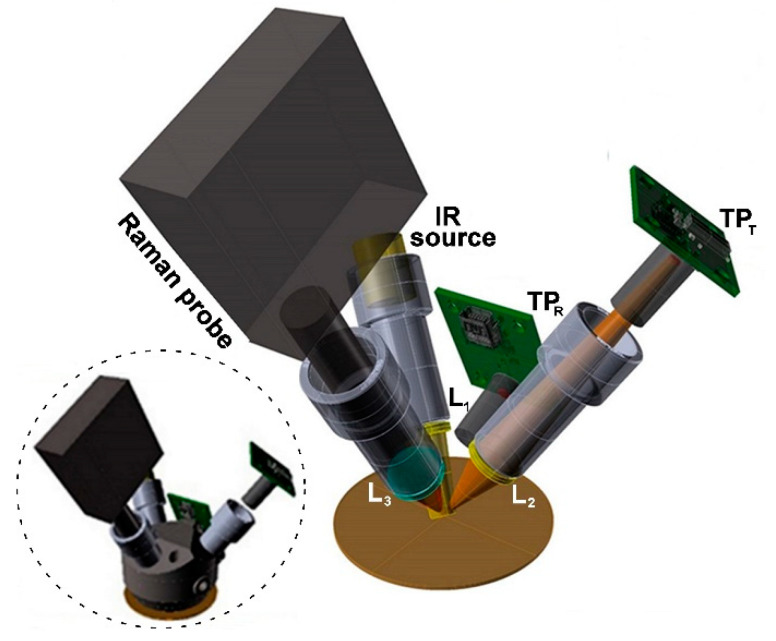
Schematic setup of the thermally controlled Raman probe including emissivity and temperature measurements channels (see the text). The four radiation channels were housed in the integrating sphere shown in the inset.

**Figure 2 sensors-22-02680-f002:**
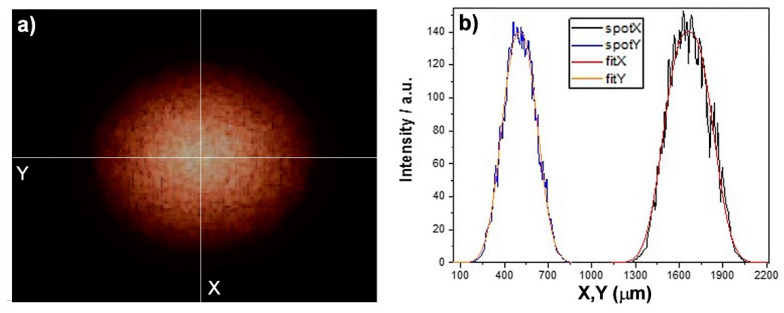
(**a**) Image of the focal spot of the laser excitation beam (500×). (**b**) Transverse (blue line) and longitudinal (black line) section of the focal spot and corresponding best fit (orange and red line) using super-Gaussian functions.

**Figure 3 sensors-22-02680-f003:**
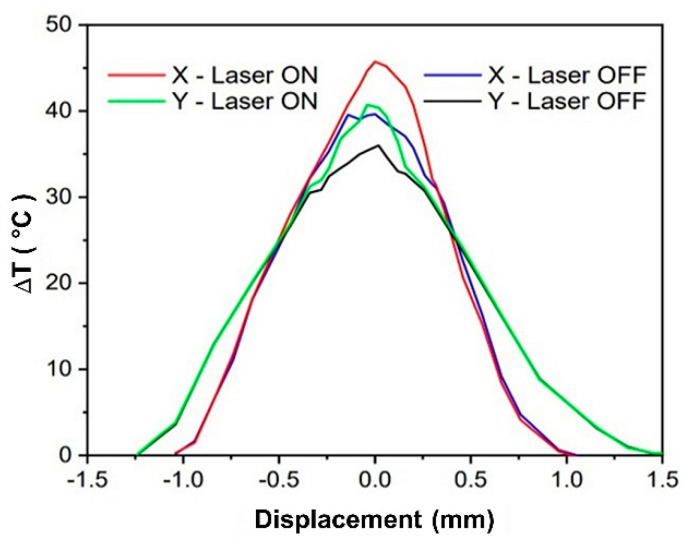
Temperature spatial profiles, as detected by the TP_T_ sensor during Y scan of a hot wire in the focal plane with laser OFF (black) and ON (green), and along X scan with laser OFF (blue) and ON (red).

**Figure 4 sensors-22-02680-f004:**
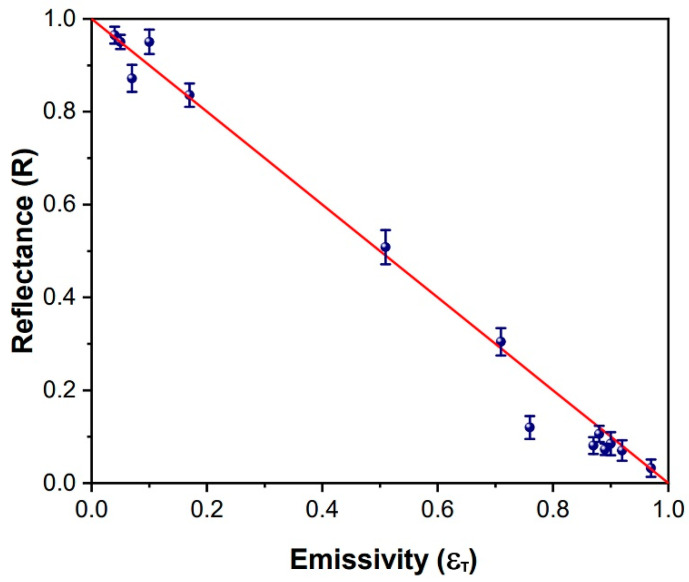
Plot comparing the (*R*, *ε_T_*) data of Table 1 and the expected linear dependence *R* = 1 − *ε*.

**Figure 5 sensors-22-02680-f005:**
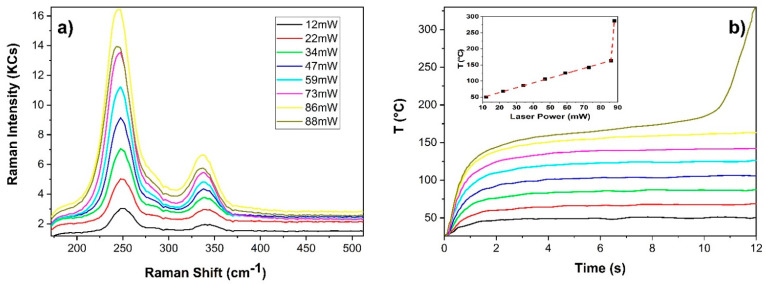
Raman spectra (**a**) obtained for cinnabar powder at different laser power and corresponding temperature temporal profiles (**b**). In the inset, the trend of temperature measured at 11.6 s as function of the laser power.

**Figure 6 sensors-22-02680-f006:**
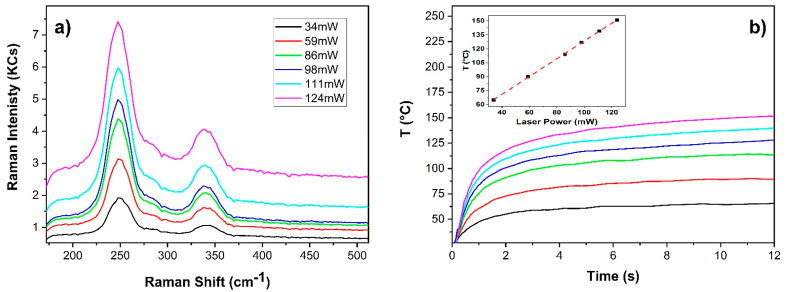
Cinnabar in linseed oil paint layer: (**a**) Raman spectra (exc. wav. 785 nm) achieved with 12 s integration time at different laser power; (**b**) corresponding temperature profiles and temperature dependence on power at 11.6 s (inset).

**Figure 7 sensors-22-02680-f007:**
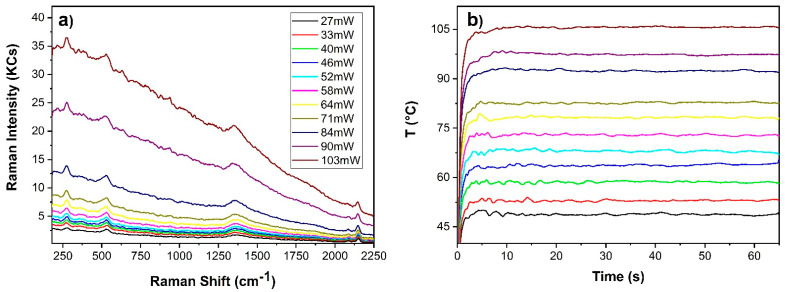
Prussian blue in linseed oil paint film: (**a**) smoothed Raman spectra collected with a single scan and integration time of 65 s; (**b**) associated temperature profiles. The emissivity value measured in situ was 0.97.

**Figure 8 sensors-22-02680-f008:**
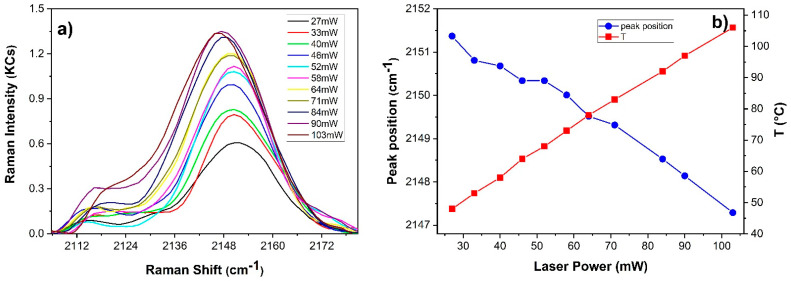
(**a**) Spectral detail of the CN stretching band (≈2150 cm^−1^) at increasing power; (**b**) Corresponding variation of the central wavenumber and temperature after 60 s laser irradiation.

**Figure 9 sensors-22-02680-f009:**
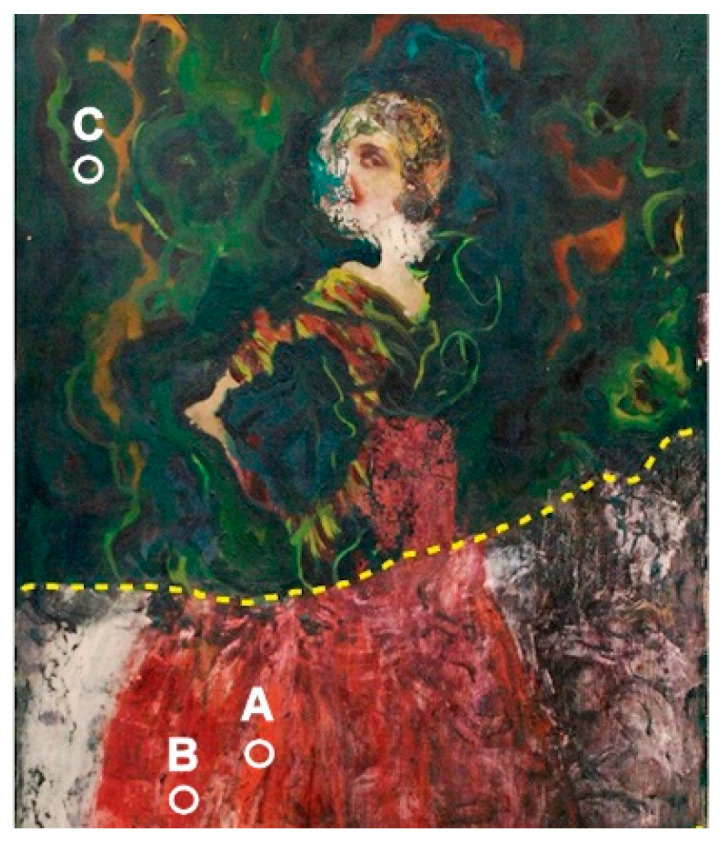
Overpainted modern oil painting by anonymous.

**Figure 10 sensors-22-02680-f010:**
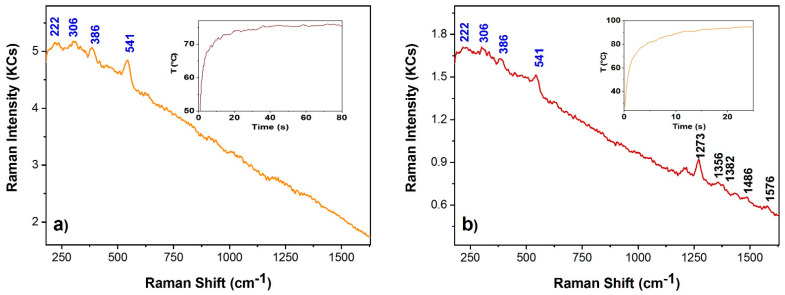
Raman spectra (exc. wav. 785 nm) of light red ((**a**) orange line) and red ((**b**) wine line) shaded zones (sites A and B, respectively) using 47 mW (**a**) and 22 mW (**b**), respectively. Blue wavenumbers correspond to red lead bands while black wavenumbers highlight the quinacridone bands. The insets show the corresponding temperature profiles, based on measured emissivity of 0.91 (**a**) and 0.87 (**b**), respectively.

**Figure 11 sensors-22-02680-f011:**
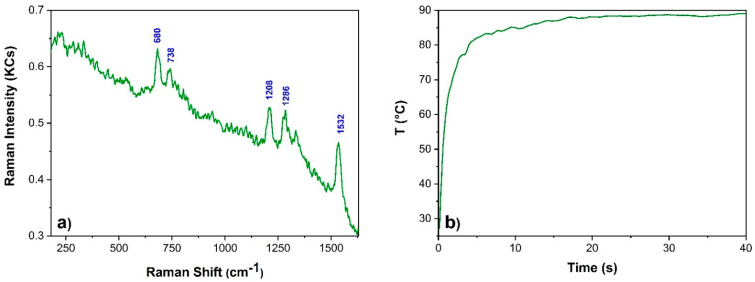
(**a**) Figure shows phthalocyanine Raman spectrum collected from the site C using 22 mW; (**b**) Temperature profile based on a measured emissivity of 0.92.

**Table 1 sensors-22-02680-t001:** Emissivity of various material samples, as measured according to the direct (*ε**_T_*) and indirect (*ε**_R_*) method, respectively, and corresponding literature data (*ε**_ref_*).

Material	*ε* _T_	*ε*_R_ = 1 − R	*ε* _ref_
Al, polished	0.04	0.04	0.04–0.06 [23]
Cu, burnished	0.05	0.05	0.07 [23]
Pb, shiny	0.07	0.13	0.08 [23]
Fe, unoxidized	0.1	0.05	0.05 [24]
AISI 316	0.17	0.16	0.14 [25]
Pb, rough	0.51	0.49	0.43 [24]
Cu, oxidized	0.71	0.70	0.60–0.70 [23]
Plywood	0.76	0.88	0.83 [26]
Gypsum	0.87	0.92	0.80–0.90 [23]
Glass, smooth	0.88	0.89	0.92–0.94 [24]
White marble	0.89	0.93	0.95 [24]
White paper	0.90	0.91	0.70–0.90 [23]
Black tape	0.92	0.93	0.90 [27]
Limestone	0.97	0.97	0.88–0.95 [24]

**Table 2 sensors-22-02680-t002:** Emissivity of pigments (powder) and paint layers with linseed oil binder using the direct (*ε_T_*) and indirect (*ε**_R_* = 1 − *R*) method, respectively.

Pigment	*ε* _T_	*ε*_R_ = 1 − R
Cinnabar	0.55	0.69
Massicot	0.63	0.53
Pb red	0.67	0.76
Cd yellow	0.79	0.90
Co blue	0.80	0.83
Cr green	0.84	0.86
Ti white	0.87	0.90
Zn white	0.84	0.88
**Paint layer**		
Linseed oil	0.88	0.96
Cinnabar	0.85	0.83
Pb red	0.93	0.93
Cd yellow	0.87	0.91
Co blue	0.92	0.97
Cr green	0.90	0.96
Ti white	0.90	0.96
Zn white	0.89	0.95

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
