# Peer review of "Novel Probe for Thermally Controlled Raman Spectroscopy Using Online IR Sensing and Emissivity Measurements"

_sensors, 2022, doi:10.3390/s22072680_

Round 1

Reviewer 2 Report

This paper calls: "Novel probe for thermally controlled Raman spectroscopy using online IR sensing and emissivity measurements" and concern of novel methods of Ramam spectroscopy for materials identification based on combination surfcace emission and infrared emission. The advantage of article is actual theme or researching and significant references review (34 ref). The disadvantege of this article are problems with formating e.g.:

Fig. 1. - absence of text about figure 1. (no desciption of experimental setup)

Fig  11 - picture gap.

Author Response

Response to the Reviewer 2

  1. The disadvantege of this article are problems with formating e.g.:

Fig. 1. - absence of text about figure 1. (no desciption of experimental setup)

Fig 11 - picture gap.

- Corrected.
